# Feature Learning in $L_2$-regularized DNNs: Attraction/Repulsion and Sparsity

**Arthur Jacot**
Courant Institute of Mathematical Sciences
New York University
arthur.jacot@nyu.edu

**Eugene Golikov**
Chair of Statistical Field Theory
École Polytechnique Fédérale de Lausanne
evgenii.golikov@epfl.ch

**Clément Hongler**
Chair of Statistical Field Theory
École Polytechnique Fédérale de Lausanne
clement.hongler@epfl.ch

**Franck Gabriel**
Institut de Science Financière et d'Assurances
Université Lyon 1
franckr.gabriel@gmail.com

## Abstract

We study the loss surface of DNNs with $L_2$ regularization. We show that the loss in terms of the parameters can be reformulated into a loss in terms of the layerwise activations $Z_\ell$ of the training set. This reformulation reveals the dynamics behind feature learning: each hidden representations $Z_\ell$ are optimal w.r.t. to an attraction/repulsion problem and interpolate between the input and output representations, keeping as little information from the input as necessary to construct the activation of the next layer. For positively homogeneous non-linearities, the loss can be further reformulated in terms of the covariances of the hidden representations, which takes the form of a partially convex optimization over a convex cone.

This second reformulation allows us to prove a sparsity result for homogeneous DNNs: any local minimum of the $L_2$-regularized loss can be achieved with at most $N(N+1)$ neurons in each hidden layer (where $N$ is the size of the training set). We show that this bound is tight by giving an example of a local minimum that requires $N^2/4$ hidden neurons. But we also observe numerically that in more traditional settings much less than $N^2$ neurons are required to reach the minima.

## 1 Introduction

It is generally believed that the success of deep learning hinges on the ability of deep neural networks (DNNs) to learn features that are well suited to the task they are trained on. There is however little understanding of what these features are and how they are selected by the network.

On the other hand, recent results [12] have shown that it is possible to train DNNs without feature learning. This suggests the existence of two regimes of DNNs, a kernel regime (also called lazy or NTK regime) without feature learning and an active regime where features are learned. The presence or absence of feature learning can depend on multiple factors, such as the initialization/parametrization of DNNs [3, 25, 17, 13], very large depths [10] or large learning rate [16, 5].

In this paper, we focus on the impact of $L_2$ regularization on feature learning in DNNs. This analysis is further motivated by recent results [8, 9, 4] which show that the implicit bias of gradient descent on losses such as the cross entropy (which decay exponentially towards infinity) is essentially the same as the bias induced by $L_2$ regularization in DNNs.

36th Conference on Neural Information Processing Systems (NeurIPS 2022).

Generally, the bias induced by the addition of $L_2$-regularization on the parameters $\theta$ of a model $f_\theta$ can be described by the *representation cost* $R(f) = \min_{\theta:f_\theta = f} \|\theta\|^2$, since $\min_\theta C(f_\theta) + \lambda \|\theta\|^2 = \min_f C(f) + \lambda R(f)$.

In deep linear networks, the addition of $L_2$ regularization on the parameters corresponds to the addition of an $L_p$-Schatten norm regularization to the represented matrix, with $p = 2/L$ where $L$ is the depth of the network [9, 6]. This implies a sparsity effect that increases with depth $L$.

In non-linear networks the sparsity effect of $L_2$-regularization has been described for shallow networks ($L = 2$) in [1, 20, 23] or for shallow non-linear networks with added linear layers [19]. Though it seems natural that this effect should become stronger for deeper networks, to our knowledge little theoretical work has been done in this area.

## 1.1 Contributions

In this paper, we study the minima of the loss of $L_2$ regularized fully-connected DNNs of depth $L$. We propose two reformulations of the loss:

1. The first reformulation expresses the loss in terms of the representations $Z_1, \ldots, Z_L$ (the layer pre-activations of every input in the training set) of every layer of the network. This reformulation has the advantage of being local - the optimal choice of a layer $Z_\ell$ only depends on its neighboring layers $Z_{\ell-1}$ and $Z_{\ell+1}$. The optimal choice of representation $Z_\ell$ is at the balance between an attractive force (determined by the previous layer) and a repulsive force (coming from the next layer). It illustrates how the representations $Z_1, \ldots, Z_{L-1}$ interpolates between the input layer $Z_0$ and output layer $Z_L$.

2. The second reformulation expresses the loss in terms of the covariances of the representation before applying the non-linearity $K_\ell = Z_\ell^T Z_\ell$ and after the non-linearity $K_\ell^\sigma = (Z_\ell^\sigma)^T Z_\ell^\sigma$. For positively homogeneous non-linearities and when the number of neurons $n_\ell$ in every hidden layer $\ell$ is larger or equal to $N(N+1)$ for $N$ the number of datapoints, this reformulation is an optimization of a (partially convex) loss over the covariances $(K_1, K_1^\sigma), \ldots, (K_{L-1}, K_{L-1}^\sigma)$ and the outputs $Z_L$, restricted to a (translated) convex cone. This reformulation does not depend on the number of neurons $n_\ell$ in the hidden layers (as long as $n_\ell \geq N(N+1)$).

The second reformulation implies that for positively homogeneous non-linearities such as the ReLU, as the number of neurons in the hidden layers $n_\ell$ increase, the global minimum of the $L_2$-regularized loss goes down until reaching a plateau (i.e. adding neurons does not lead to an improvement in the loss). This illustrates the sparsity effect of $L_2$-regularization, where the optimum reached on a very large network is equivalent to a much smaller network.

The start of the plateau hence gives a measure of sparsity of the global minimum. We show that the minimal number of neurons $n_\ell$ to reach this plateau is determined by a notion of rank $\mathrm{Rank}_\sigma (K_\ell, K_\ell^\sigma)$ of the covariance pairs. We show that $\mathrm{Rank}_\sigma (K_\ell, K_\ell^\sigma) \leq N(N+1)$, i.e. the plateau must start before $N(N+1)$ and that the scaling of this upper bound is tight by giving an example dataset such that at the optimum $\mathrm{Rank}_\sigma (K_\ell, K_\ell^\sigma) \geq N^2/4$. We also present other datasets where the start of the plateau is either constant or grows linearly with the number of datapoints. We also observe empirically that the plateau can start at much smaller widths for real data such as MNIST and the teacher/student setting.

## 2 Setup

We consider fully-connected deep neural networks with $L+1$ layers, numbered from $0$ (the input layer) to $L$ (the output layer), with nonlinear activation function $\sigma : \mathbb{R} \to \mathbb{R}$ (e.g. the ReLU $\sigma(x) = \max\{0, x\}$)). Each layer $\ell$ contains $n_\ell$ neurons and we denote $\mathbf{n} = (n_1, \ldots, n_L)$ the widths of the network. Given an input dataset $\{x_1, \ldots, x_N\} \subset \mathbb{R}^{n_0}$ of size $N$, we consider the data matrix $X = (x_0, \ldots, x_n) \in \mathbb{R}^{n_0 \times N}$, and encode the activations and preactivations of the whole data set by considering the pre-activations $Z_\ell(X; \mathbf{W}) \in \mathbb{R}^{n_\ell \times N}$ and activations $Z_\ell^\sigma(X; \mathbf{W}) \in \mathbb{R}^{(n_\ell+1) \times N}$

given by:

$$Z_0^\sigma(X; \mathbf{W}) = \begin{pmatrix} X \\ \beta \mathbf{1}_N^T \end{pmatrix}$$

$$Z_\ell(X; \mathbf{W}) = W_\ell Z_{\ell-1}^\sigma(X; \mathbf{W})$$

$$Z_\ell^\sigma(X; \mathbf{W}) = \begin{pmatrix} \sigma(Z_\ell) \\ \beta \mathbf{1}_N^T \end{pmatrix},$$

where $\mathbf{W} = (W_\ell)_{\ell=1,\ldots,L}$ is the collection of $n_\ell \times (n_{\ell-1} + 1)$-dim weight matrices $W_\ell$, $\sigma(Z_\ell)$ is obtained by applying elementwise the nonlinearity $\sigma$ to the matrix $Z_\ell$, and the scalar $\beta \in \mathbb{R}$ represents the amount of bias (i.e. when $\beta = 0$ there is no bias, when $\beta = 1$ this definition is equivalent to the traditional definition of bias). The output of the network is the pre-activation of the $L$-th layer $Z_L$.

We often drop the dependence on the weights $\mathbf{W}$ and on the dataset $X$ and simply write $Z_\ell$ and $Z_\ell^\sigma$.

We denote $f_\mathbf{W} : \mathbb{R}^{n_0} \to \mathbb{R}^{n_L}$ the *network function*, which maps an input $x$ to the pre-activation at the last layer.

### 2.1 $L_2$-Regularized Loss and Representation Cost

Given a general cost functional $C : \mathbb{R}^{n_L \times N} \to \mathbb{R}$, the $L_2$-regularized loss of DNNs of widths $\mathbf{n}$ is

$$\mathcal{L}_{\lambda,\mathbf{n}}(\mathbf{W}) = C(Z_L(X; \mathbf{W})) + \lambda \|\mathbf{W}\|^2,$$

where $\|\mathbf{W}\|$ is the $L_2$-norm of $\mathbf{W}$ understood as a vector. Note that $\|\mathbf{W}\|^2 = \sum_{\ell=1}^L \|W_\ell\|_F^2$ where $\|\cdot\|_F$ denotes the Frobenius norm. From now on, we often omit to specify the widths $\mathbf{n}$ and simply write $\mathcal{L}_\lambda$.

The additional regularization cost should bias the network toward low norm solutions. This bias on the parameters leads to a bias in function space, which is described by the so-called *representation cost* $\mathcal{R}_\mathbf{n}(f)$ defined on functions $f : \mathbb{R}^{n_0} \to \mathbb{R}^{n_L}$:

$$\mathcal{R}_\mathbf{n}(f) = \min_{\mathbf{W}:f_\mathbf{W}=f} \|\mathbf{W}\|^2,$$

where the minimum is taken over all choices of parameters $\mathbf{W}$ of a width $\mathbf{n}$ network, with fixed $\beta$ bias amount, such that the network function $f_\mathbf{W}$ equals $f$. By convention, if no such parameters exist then $\mathcal{R}_\mathbf{n}(f) = +\infty$.

Similarly, given an input-output pair $X \in \mathbb{R}^{n_0 \times N}$, $Y \in \mathbb{R}^{n_L \times N}$, the representation cost $R_\mathbf{n}(X, Y)$ is:

$$R_\mathbf{n}(X, Y) = \min_{\mathbf{W}:Z_L(X,\mathbf{W})=Y} \|\mathbf{W}\|^2,$$

with again the convention that if there exists no weight $\mathbf{W}$ such that $Z_L(X, \mathbf{W}) = Y$, then $R_\mathbf{n}(X, Y) = +\infty$. The representation cost $R_\mathbf{n}$ naturally describes the bias induced by the $L_2$-regularized loss of DNNs since:

$$\min_\mathbf{W} C(Z_L(X; \mathbf{W})) + \lambda \|\mathbf{W}\|^2 = \min_Y C(Y) + \lambda R_\mathbf{n}(Y, X).$$

## 3 Two Reformulations of the Regularized Loss: Hidden Representation and Covariance Optimization

We now provide two reformulations of the $L_2$-regularized loss $\mathcal{L}_\lambda(\mathbf{W})$ and representation cost $R_\mathbf{n}(X, Y)$, which both put emphasis on the hidden representations $Z_\ell$ and how they are progressively modified throughout the neural network. The first reformulation holds for general non-linearities while the second only applies to networks with homogeneous nonlinearities.

### 3.1 Feature optimization : attraction/repulsion

The key observation is that the weights $W_\ell$ can be decomposed as follows:

$$W_\ell = Z_\ell \left(Z_{\ell-1}^\sigma\right)^+ + \tilde{W}_\ell,$$

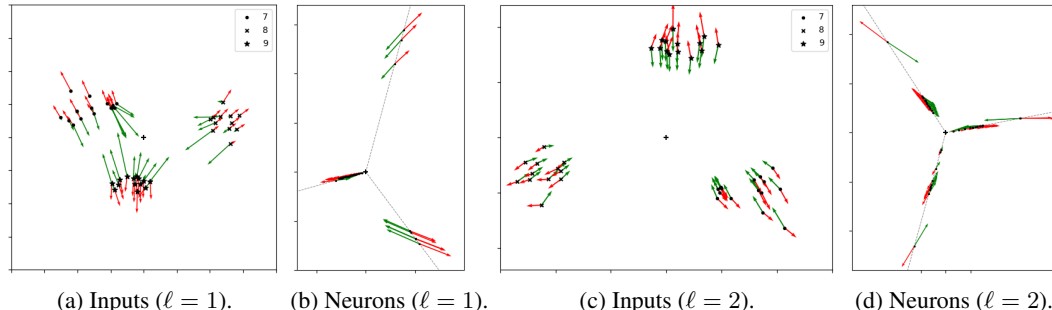

| (a) Inputs ($\ell = 1$). | (b) Neurons ($\ell = 1$). | (c) Inputs ($\ell = 2$). | (d) Neurons ($\ell = 2$). |

Figure 1: **Attraction/Repulsion:** Visualization of the hidden representation $Z_1$ and $Z_2$ of a $L = 3$ ReLU DNN at the end of training (i.e. after $T = 20k$ steps of gradient descent on the original loss $\mathcal{L}_\lambda$) on 3 digits (7,8 and 9) of MNIST [15] along with the attraction force in green and repulsion force in red (both forces are approximated with Tikhonov regularization). For both layers, we plot in **(a)** and **(c)** the PCA of the $N = 42$ lines (each corresponding to a datapoint) and in **(b)** and **(d)** the PCA the $n_1 = n_2 = 50$ columns (each corresponding to a neuron in the layer $\ell = 1$ or $\ell = 2$). We observe a clustering of the inputs according to their digit, and of the neurons along 3 rays in grey dashed lines.

where the residual matrix $\tilde{W}_\ell$ is orthogonal to $Z_{\ell-1}^\sigma$, i.e. $\tilde{W}_\ell Z_{\ell-1}^\sigma = 0$, and $(\cdot)^+$ is the Moore-Penrose pseudo-inverse. This stems from the fact that $W_\ell = W_\ell \mathrm{P}_{\mathrm{Im} Z_{\ell-1}^\sigma} + \tilde{W}_\ell$ where $\tilde{W}_\ell := W_\ell \mathrm{P}_{(\mathrm{Im} Z_{\ell-1}^\sigma)^\perp}$, and $\mathrm{P}_{\mathrm{Im} Z_{\ell-1}^\sigma}$, resp. $\mathrm{P}_{(\mathrm{Im} Z_{\ell-1}^\sigma)^\perp}$, is the orthogonal projection on $\mathrm{Im} Z_{\ell-1}^\sigma$, resp. on the orthogonal complement of $\mathrm{Im} Z_{\ell-1}^\sigma$; one concludes using the facts that $\mathrm{P}_{\mathrm{Im} Z_{\ell-1}^\sigma} = Z_{\ell-1}^\sigma \left(Z_{\ell-1}^\sigma\right)^+$ and $Z_\ell = W_\ell Z_{\ell-1}^\sigma$.

Note that the matrix $\tilde{W}_\ell$ does not affect either the hidden representations $Z_\ell$ nor the output $Z_L$. Besides, the Frobenius norm of $W_\ell$ can be rewritten as $\|W_\ell\|_F^2 = \|Z_\ell \left(Z_{\ell-1}^\sigma\right)^+\|_F^2 + \|\tilde{W}_\ell\|_F^2$. When minimizing the $L_2$-regularized cost, it is therefore always optimal to consider null residual matrices $\tilde{W}_\ell = 0$, resulting in a reformulation of the cost which only depends on the pre-activations $Z_\ell$:

**Proposition 1.** *The infimum of $\mathcal{L}_\lambda(\mathbf{W}) = C(Z_L(X; \mathbf{W})) + \lambda \|\mathbf{W}\|^2$, over the parameters $\mathrm{W} \in \mathbb{R}^P$ is equal to the infimum of*

$$\mathcal{L}_\lambda^r(Z_1, \ldots, Z_L) = C(Z_L) + \lambda \sum_{\ell=1}^L \left\| Z_\ell \left(Z_{\ell-1}^\sigma\right)^+ \right\|_F^2$$

*over the set $\mathcal{Z}$ of hidden representations $\mathbf{Z} = (Z_\ell)_{\ell=1,\ldots,L}$ such that $Z_\ell \in \mathbb{R}^{n_\ell \times N}$, $\mathrm{Im} Z_{\ell+1}^T \subset \mathrm{Im}\left(Z_\ell^\sigma\right)^T$, with the notations $Z_0^\sigma = \begin{pmatrix} X \\ \beta \mathbf{1}_N^T \end{pmatrix}$ and $Z_\ell^\sigma = \begin{pmatrix} \sigma(Z_\ell) \\ \beta \mathbf{1}_N^T \end{pmatrix}$.*

*Furthermore, if $\mathbf{W}$ is a local minimizer of $\mathcal{L}_\lambda$ then $(Z_1(X; \mathbf{W}), \ldots, Z_L(X; \mathbf{W}))$ is a local minimizer of $\mathcal{L}_\lambda^r$. Conversely, keeping the same notations, if $(Z_\ell)_{\ell=1,\ldots,L}$ is a local minimizer of $\mathcal{L}_\lambda^r$, then $\mathrm{W} = (Z_\ell(Z_{\ell-1}^\sigma)^+)_{\ell=1,\ldots,L}$ is a local minimizer of $\mathcal{L}_\lambda$.*

Note that one can also reformulate the representation cost:

$$R_{\mathbf{n}}(X, Y) = \min_{\mathbf{Z} \in \mathcal{Z}, Z_L = Y} \sum_{\ell=1}^L \left\| Z_\ell \left(Z_{\ell-1}^\sigma\right)^+ \right\|_F^2.$$

The representation in terms of the output and hidden representations have several interesting properties, especially when it comes to minimization:

1. The optimization becomes local in the sense that all terms and constraints depend either only on the output cost $C(Z_L)$ or on two neighboring terms (e.g. $\left\| Z_\ell \left(Z_{\ell-1}^\sigma\right)^+ \right\|_F^2$). As a result, the (projected) gradient of the loss $\mathcal{L}_\lambda^r(Z_1, ..., Z_L)$ w.r.t. to $Z_\ell$ only depends on $Z_{\ell-1}, Z_\ell$ and $Z_{\ell+1}$. This is in contrast to the optimization of $\mathcal{L}_\lambda(\mathbf{W})$, where the gradient of $C(Z_L)$ with respect to $W_\ell$ depends on all parameters $W_1, \ldots, W_L$.

2. The value $\left\| Z_\ell \left( Z_{\ell-1}^\sigma \right)^+ \right\|_F^2$ represents a 'multiplicative distance' between $Z_\ell$ and $Z_{\ell-1}^\sigma$ (in contrast to the 'additive distance' $\left\| Z_\ell - Z_{\ell-1}^\sigma \right\|_F^2$); the representation $Z_\ell$ therefore interpolates multiplicatively between $Z_{\ell-1}$ and $Z_{\ell+1}$. This is most obvious for linear networks (i.e. $\sigma = \mathrm{id}$ and $\beta = 0$): in this case, one can check that at any global minimizer, the covariances of the hidden layers equal $Z_\ell^T Z_\ell = X^T (X^{-T} Z_L^T Z_L X^{-1})^{\frac{\ell}{L}} X$, interpolating between the input covariance $X^T X$ and output covariance $Z_L^T Z_L$.

3. A lot of work has been done to propose biologically plausible training methods for DNNs [2, 11], in contrast to backpropagation which is not local. A line of work [21, 18, 22], propose a biologically plausible optimization technique which minimizes a cost which closely resembles our first reformulation, with the multiplicative distances $\left\| Z_\ell \left( Z_{\ell-1}^\sigma \right)^+ \right\|_F^2$ replaced by additive ones $\left\| Z_\ell - Z_{\ell-1}^\sigma \right\|_F^2$. Due to this change, there is no direct correspondence between the networks trained with this biologically plausible technique and those trained with backpropagation. If one could extend this training technique to work with multiplicative distances one could guarantee such a direct correspondence.

4. The optimization leads to an attraction-repulsion algorithm. If we optimize only on the term $Z_\ell$ and fix all other representations, the only two terms that depend on $Z_\ell$ are $\left\| Z_\ell \left( Z_{\ell-1}^\sigma \right)^+ \right\|_F^2$ and $\left\| Z_{\ell+1} \left( Z_\ell^\sigma \right)^+ \right\|_F^2$. The former term is attractive as it pushes the representations $Z_\ell$ towards the origin (and hence pushes the representations at depth $\ell$ of every input towards each other), especially along directions where $Z_{\ell-1}^\sigma$ is small. The latter term is repulsive as it pushes the representations $Z_\ell^\sigma$ away from the origin, especially along directions where $Z_{\ell+1}$ is large.

5. This attraction-repulsion process is similar to the Information Bottleneck theory [24]: the repulsive term ensure that $Z_\ell$ keeps enough information about the inputs to reconstruct $Z_{\ell+1}$, while the attractive term pushes $Z_\ell$ to keep as little information as possible.

The attraction and repulsion forces of of the $\ell$-th layer are the derivative $\partial_{Z_\ell} \left\| Z_\ell \left( Z_{\ell-1}^\sigma \right)^+ \right\|_F^2$ and $\partial_{Z_\ell} \left\| Z_{\ell+1} \left( Z_\ell^\sigma \right)^+ \right\|_F^2$ which are both $n_\ell \times N$ matrices. One can visualize these forces either column by column (each column corresponding to a datapoint $i = 1, \ldots, N$) or line by line (each line corresponding to a neuron $k = 1, \ldots, n_\ell$). These two visualizations of the forces are presented in Figure 1 for the two hidden layers of a depth $L = 3$ network, projected to the 2 largest principal components of the columns resp. lines of $Z_\ell$. Figures 1a and 1c illustrate how the inputs corresponding to different classes are pushed away from each other, leading to a clustering effect. Figures 1b and 1d show that the neurons naturally align along rays starting from the origin. This happens for homogeneous non-linearities, such as the ReLU in this example, because if two neurons $k, k'$ have proportional activations, i.e. $Z_{\ell,k} = \alpha Z_{\ell,k}$ for some $\alpha \in \mathbb{R}$, then their attractive and repulsive forces will also be proportional with the same scaling $\alpha$. As a result, the neuron $k$ is stable, i.e. the attraction and repulsion cancel each other, if and only if the neuron $k'$ is stable.

This phenomenon can be interpreted as a form of sparsity: a group of aligned neurons can be replaced by a single neuron without changing the resulting function $Z_L$. Can we guarantee a degree of sparsity in the hidden representations? Can we bound the number of aligned groups in a neuron? In the next section, we introduce a further reformulation of the loss which allows us to partially answer these questions.

### 3.2 Covariance learning : partial convex optimization for positively homogeneous nonlinearities

The loss of the first reformulation $\mathcal{L}_\lambda^r$ depends on the hidden representations $Z_\ell$ and $Z_\ell^\sigma$ only through the covariances $K_\ell = Z_\ell^T Z_\ell$ and $K_\ell^\sigma = \left( Z_\ell^\sigma \right)^T Z_\ell^\sigma$, since $\left\| Z_\ell \left( Z_{\ell-1}^\sigma \right)^+ \right\|_F^2 = \mathrm{Tr}\left[ K_\ell \left( K_{\ell-1}^\sigma \right)^+ \right]$. Hence, we provide a second reformulation expressed in terms of the tuple of covariance pairs $\mathbf{K} = ((K_1, K_1^\sigma), \ldots, (K_{L-1}, K_{L-1}^\sigma))$ and the outputs $Z_L$. Using the notations $K_0^\sigma = X^T X + \beta^2 \mathbf{1}_{N \times N}$

and $K_L = Z_L^T Z_L$, we define:

$$\mathcal{L}_\lambda^k(\mathbf{K}, Z_L) = C(Z_L) + \lambda \sum_{\ell=1}^{L} \text{Tr}\left[K_\ell \left(K_{\ell-1}^\sigma\right)^+\right].$$

It remains to identify the set $\mathcal{K}_{\mathbf{n}}(X)$ of covariances $\mathbf{K}$ and outputs $Z_L$ which can be represented by a width $\mathbf{n}$ network with inputs $X$. For positively homogeneous nonlinearities of degree 1 such as the ReLU (i.e. when $\sigma(\lambda x) = \lambda \sigma(x)$ for any positive $\lambda$), the set $\mathcal{K}_{\mathbf{n}}(X)$ can be expressed using the notion of conical hulls.

**Definition 2.** The conical hull of $\Omega \subset \mathbb{R}^d$ is the set $\text{cone}(\Omega) := \left\{\sum_{i=1}^{k} \alpha_i \omega_i : k \geq 0, \alpha_i \geq 0, \omega_i \in \Omega\right\}$ and its $m$-conical hull for $m \geq 1$ is the set $\text{cone}_m(\Omega) := \left\{\sum_{i=1}^{m} \alpha_i \omega_i : \alpha_i \geq 0, \omega_i \in \Omega\right\}$.

Note that by Caratheodory's theorem for conical hulls, $\text{cone}_m(\Omega) = \text{cone}(\Omega)$ for any $m \geq d$. We now proceed to the description of the set $\mathcal{K}_{\mathbf{n}}(X)$ and obtain the second formulation of the $L_2$ regularized loss and of the representation loss $R_n$ in terms of covariances:

**Proposition 3.** *For positively homogeneous non-linearities $\sigma$, the infimum of $\mathcal{L}_\lambda(\mathbf{W}) = C(Z_L(X; \mathbf{W})) + \lambda \|\mathbf{W}\|^2$, over the parameters $\mathbf{W} \in \mathbb{R}^P$ is equal to the infimum over $\mathcal{K}_{\mathbf{n}}(X)$ of*

$$\mathcal{L}_\lambda^k(\mathbf{K}, Z_L) = C(Z_L) + \lambda \sum_{\ell=1}^{L} \text{Tr}\left[K_\ell \left(K_{\ell-1}^\sigma\right)^+\right].$$

*The set $\mathcal{K}_{\mathbf{n}}(X)$ is the set of covariances $\mathbf{K} = ((K_1, K_1^\sigma), \ldots, (K_{L-1}, K_{L-1}^\sigma))$ and outputs $Z_L$ such that for all hidden layer $\ell = 1, \ldots, L-1$:*

- *the pair $(K_\ell, K_\ell^\sigma)$ belongs to the (translated) $n_\ell$-conical hull*

$$S_{n_\ell} = \text{cone}_{n_\ell}\left(\left\{\left(xx^T, \sigma(x)\sigma(x)^T\right) : x \in \mathbb{R}^N\right\}\right) + (0, \beta^2 \mathbf{1}_{N \times N}),$$

- $\text{Im} K_\ell \subset \text{Im} K_{\ell-1}^\sigma$, *with the notation $K_0^\sigma = X^T X + \beta^2 \mathbf{1}_{N \times N}$, and $\text{Im} Z_L \subset \text{Im} K_{L-1}^\sigma$.*

Note that one can also reformulate the representation cost:

$$R_{\mathbf{n}}(X, Y) = \min_{\mathbf{K}:(\mathbf{K}, Y) \in \mathcal{K}_{\mathbf{n}}(X)} \sum_{\ell=1}^{L} \text{Tr}\left[K_\ell \left(K_{\ell-1}^\sigma\right)^+\right].$$

*Remark* 4. In contrast to the first loss $\mathcal{L}_\lambda^r$ whose local minima were in correspondence with the local minima of the original loss $\mathcal{L}_\lambda$, the second loss $\mathcal{L}_\lambda^k$ can in some cases have strictly less critical points and local minima. Indeed, since the map $\mathbf{W} \mapsto (\mathbf{K}, Z_L)$ is continuous, if $(\mathbf{K}(\mathbf{W}), Z_L(\mathbf{W}))$ is a local minimum, then so is $\mathbf{W}$. However, the converse is not true: we provide a counterexample in the Appendix, i.e. a set of weights $\mathbf{W}$ of a depth $L = 2$ network which is a local minimum of $\mathcal{L}_\lambda$ and such that the corresponding $(\mathbf{K}, Z_L)$ is not a local minimum of $\mathcal{L}_\lambda^k$.

Since the dimension of the space of pairs of symmetric $N \times N$ matrices is $N(N+1)$, if $n_\ell \geq N(N+1)$ then, by the Caratheodory's theorem for conical hulls,

$$S_{n_\ell} = S := \text{cone}\left(\left\{\left(xx^T, \sigma(x)\sigma(x)^T\right) : x \in \mathbb{S}^{N-1}\right\}\right) + (0, \beta^2 \mathbf{1}_{N \times N}).$$

Hence, as soon as $n_\ell \geq N(N+1)$ for all hidden layers, the set $\mathcal{K}_{\mathbf{n}}(X)$ does not depend on the list of widths $\mathbf{n}$. We denote

$$\mathcal{K}(X) = \left\{(\mathbf{K}, Z_L) \mid \forall \ell = 1, \ldots, L-1, (K_\ell, K_\ell^\sigma) \in S, \text{Im} K_\ell \subset \text{Im} K_{\ell-1}^\sigma, \text{Im} Z_L \subset \text{Im} K_{L-1}^\sigma\right\}$$

this width-independent set. The following proposition shows that for sufficiently wide networks with a positively homogeneous nonlinearity, training a deep DNN with $L_2$ regularization is equivalent to a partially convex optimization over a translated convex cone.

**Proposition 5.** *The set $\mathcal{K}(X)$ is a translated convex cone: after the suitable translation, it is equal to its conical hull. The cost $\mathcal{L}_\lambda^k(\mathbf{K}, Z_L)$ is partially convex w.r.t. to the outputs $Z_L$ and the pairs $(K_\ell, K_\ell^\sigma)$, i.e. it is convex if one fixes the other parameters and let only $(K_\ell, K_\ell^\sigma)$, or $Z_L$, vary.*

## 3.3 Direct optimization of the reformulations

It is natural at this point to wonder whether one could optimize directly over the representations $\mathbf{Z}$ (using the first reformulation) or over the covariances $\mathbf{K}$ and output $Z_L$ (using the second reformulation), and whether this would have an advantage over the traditional optimization of the weights.

For the first reformulation, one can simply use the projected gradient descent with updates given by

$$\mathbf{Z}_{t+1} = P_{\mathcal{Z}} \left( \mathbf{Z}_t - \eta \nabla \mathcal{L}_\lambda^r (\mathbf{Z}) \right)$$

for any projection $P_{\mathcal{Z}}$ to the constraint space $\mathcal{Z}$. For example, a projection is obtained by mapping $Z_\ell$ to $Z_\ell P_{\mathrm{Im} Z_{\ell-1}^\sigma}$ sequentially from $\ell = 1$ to $\ell = L$. Note however that the loss explodes as the constraints become unsatisfied so that for gradient flow, there is no need for the projections. This suggests that these projections might also be unnecessary as long as the learning rate is small enough. For more details, see Appendix B.1.

For the second reformulation, there is no obvious way to compute a projection to the constraint space $\mathcal{K}$: the cone $S$ is spanned by an infinite amount of points and we do not have an explicit formula for the dual cone $S^*$. Frank-Wolfe optimization can be used to overcome the need for computing the projections.

However, these direct optimizations of the reformulations lead to issues of computational complexity and stability. First, the computation of the gradients $\nabla \mathcal{L}_\lambda^r (\mathbf{Z})$ and $\nabla \mathcal{L}_\lambda^k (\mathbf{K}, Z_L)$ requires solving a linear equation of dimension $N$, which is very costly, in contrast to the traditional optimization of the weights $\mathbf{W}$ for which the gradient can be computed very efficiently. Second, if $Z_\ell^\sigma$ is not full-rank, the computation of its pseudo-inverse $(Z_\ell^\sigma)^+$ and the projection $P_{\mathrm{Im} Z_\ell^\sigma}$ are very unstable. Therefore, if we only have finite-precision knowledge of $Z_\ell$, we cannot reliably compute $(Z_\ell^\sigma)^+$ nor $P_{\mathrm{Im} Z_\ell^\sigma}$.

Although it could be possible to solve these problems (e.g. using the Tikhonov regularization for the unstability problem) and to develop efficient algorithms to optimize both reformulations efficiently, we decided in this paper to focus on the theoretical implications of these reformulations.

## 4 Sparsity of the Regularized Optimum for Homogeneous DNNs

In this section, we assume that the non-linearity is positively homogenous. Under this assumption, the second reformulation of the loss (and of the representation cost) holds and implies the existence of a sparsity phenomenon.

First observe that as the widths $\mathbf{n}$ increase, both the global minimizer of the loss $\min_\mathbf{W} \mathcal{L}_\lambda(\mathbf{W})$ and the representation cost $R_\mathbf{n}(X, Y)$ diminish. We denote by $\mathcal{L}_{\lambda, \mathbf{n}}$ the $L_2$-regularized loss of DNNs with widths n. Recall that the depth $L$ is fixed.

**Proposition 6.** *If $\mathbf{n} \leq \mathbf{n}'$ (in the sense that $n_\ell \leq n'_\ell$ for all $\ell$ and $n_0 = n'_0$ and $n_L = n'_L$), then*

$$\min_{\mathbf{W} \in \mathbb{R}^{P_\mathbf{n}}} \mathcal{L}_{\lambda, \mathbf{n}}(\mathbf{W}) \geq \min_{\mathbf{W} \in \mathbb{R}^{P_{\mathbf{n}'}}} \mathcal{L}_{\lambda, \mathbf{n}'}(\mathbf{W})$$

*and for any $X \in \mathbb{R}^{n_0 \times N}$ and $Y \in \mathbb{R}^{n_L \times N}$, $R_\mathbf{n}(X, Y) \geq R_{\mathbf{n}'}(X, Y)$.*

*Proof.* Let us assume that the parameters $\mathbf{W}^*$ are optimal for a width $\mathbf{n}$ network, the parameters can be mapped to parameters of a wider network by adding 'dead' neurons (i.e. neurons with zero incoming and outcoming weights) without changing the network function $f_{\mathbf{W}^*}$ nor the norm of the parameters $\|W\|$. □

For DNNs with positively homogeneous nonlinearities, a direct consequence of our reformulation through the hidden covariances is that both the global minimum of the loss $\mathcal{L}_\lambda$ and the representation cost $R_\mathbf{n}(X, Y)$ plateau for any widths $\mathbf{n}$ such that $n_\ell \geq N(N + 1)$.

**Proposition 7.** *For any positively homogeneous nonlinearity $\sigma$, any widths $\mathbf{n}$ and $\mathbf{n}'$ such that $n_0 = n'_0$, $n_L = n'_L$ and for all $\ell = 1, \dots, L - 1$, $n_\ell, n'_\ell \geq N(N + 1)$, for all $\lambda > 0$, we have:*

$$\min_{\mathbf{W} \in \mathbb{R}^{P_\mathbf{n}}} \mathcal{L}_{\lambda, \mathbf{n}}(\mathbf{W}) = \min_{\mathbf{W} \in \mathbb{R}^{P_{\mathbf{n}'}}} \mathcal{L}_{\lambda, \mathbf{n}'}(\mathbf{W}).$$

*Under the same conditions $R_\mathbf{n}(X, Y) = R_{\mathbf{n}'}(X, Y)$ for any $X \in \mathbb{R}^{n_0 \times N}$ and $Y \in \mathbb{R}^{n_L \times N}$,.*

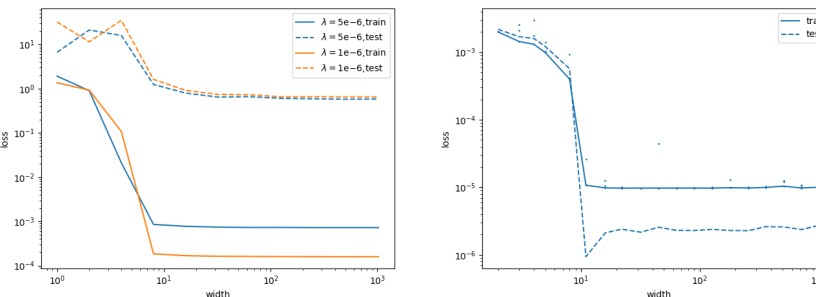

Figure 2: **Loss plateau:** Plots of the train loss (full lines) and test loss (dashed lines) as a function of the width for depth $L = 3$ DNNs for different datasets: (left) cross-entropy loss for a subset of MNIST ($N = 1000$) and two values of $\lambda$; (right) MSE with $\lambda = 10^{-6}$ on $N = 1000$ Gaussian inputs and outputs evaluated on a fixed teacher network of depth $L = 3$ and width 10( (right). For the right plot we took (the minimum is taken over 3 independent trials, represented by the small blue dots). In both settings, the plateau appears to start around 10, much earlier than $N^2 = 10^6$. The regularization term is included in the training loss but not the test, leading to a smaller test loss on the right.

*Proof.* This follows directly from the second reformulation and the fact that by Caratheodory's theorem for conical hulls, $S_n = S$ if $n \geq N(N+1)$ (see discussion after Remark 4). $\square$

We can therefore define a width-independent representation cost $R(X, Y)$ (which still depends on the fixed depth $L$) equal to the representation cost $R_\mathbf{n}(X, Y)$ of any sufficiently wide network.

### 4.1 Rank of the Hidden Representations

Now that we have revealed the plateau phenomenon, a natural question that we investigate in this section is when does this plateau begin. In order to do so, we introduce the notion of rank $\text{Rank}_\sigma(K, K^\sigma)$ of a pair of Gram matrices $(K, K^\sigma) \in S$ which is the minimal number $k$ such that

$$K = \sum_{i=1}^{k} z_i z_i^T \text{ and } K^\sigma = \sum_{i=1}^{k} \sigma(z_i) \sigma(z_i)^T + \beta^2 \mathbf{1}_{N \times N} \tag{4.1}$$

for some $z_1, \ldots, z_k \in \mathbb{R}^N$. This notion of rank describes exactly the minimal number of neurons required to recover a set of covariances $\mathbf{K}$:

**Proposition 8.** *Let $(\mathbf{K}, Z_L) \in \mathcal{K}(X)$, then there are parameters $\mathbf{W}$ of a width $\mathbf{n}$ network with covariances and outputs $\mathbf{K}$ if and only if $n_\ell \geq \text{Rank}_\sigma(K_\ell, K_\ell^\sigma)$ for all $\ell = 1, \ldots, L-1$.*

We can now describe the plateau $R = \{\mathbf{n} : \min_{\mathbf{W} \in \mathbb{R}^{P_\mathbf{n}}} \mathcal{L}_{\lambda, \mathbf{n}}(\mathbf{W}) = \min_\mathbf{m} \min_{\mathbf{W} \in \mathbb{R}^{P_\mathbf{m}}} \mathcal{L}_{\lambda, \mathbf{m}}(\mathbf{W})\}$, i.e. the set of widths $\mathbf{n}$ such that the minimum $\min_{\mathbf{W} \in \mathbb{R}^{P_\mathbf{n}}} \mathcal{L}_{\lambda, \mathbf{n}}(\mathbf{W})$ is optimal over all possible widths:

**Corollary 9.** *Let $K_{\min}$ be the set of covariances sequences $(\mathbf{K}, Z_L)$ which are global minima of the second reformulation. We have that $\mathbf{n} \in R$ if and only if there is a $(\mathbf{K}, Z_L) \in K_{\min}$ such that $n_\ell \geq \text{Rank}_\sigma(K_\ell, K_\ell^\sigma)$.*

Hence, the investigation of $\text{Rank}_\sigma(\cdot, \cdot)$ is crucial to understand where the plateau begins; unfortunately, it can be difficult to compute. However, from its definition and the Caratheodory's theorem for conical hulls (see our discussion after Remark 4), we have the following natural bounds:

**Lemma 10.** *For any pair $(K, K^\sigma) \in S$, we have $\text{Rank}(K) \leq \text{Rank}_\sigma(K, K^\sigma) \leq N(N+1)$.*

We show in the next section that the order of magnitude of the upper bound is tight. More specifically, we construct a dataset for which any global optimum satisfies $\text{Rank}_\sigma(K_1, K_1^\sigma) \geq N^2/4$. This implies that, in this example, the plateau transition occurs when the number of hidden neurons is of order $O(N^2)$. Note however that, in our numerical experiments (see Figure 2), the rank of the global optimum can be much smaller for more traditional dataset such as MNIST.

*Remark* 11. The start of the plateau measures a notion of sparsity of the learned network, since the networks learned in the plateau are equivalent to a network at the start of the plateau, i.e. large networks are equivalent in terms of their covariances and outputs $(\mathbf{K}, Z_L)$ to a (potentially much) smaller network.

Even though the set of pairs $(K_\ell, K_\ell^\sigma)$ in the cone $S$ that are not full rank has measure zero, the optimal representations $(K_\ell, K_\ell^\sigma)$ always lie on the border of the cone $S$ (since the derivative of the cost $\mathcal{L}_\lambda^k$ w.r.t. $(K_\ell, K_\ell^\sigma)$ never vanishes) where the rank is lower. More precisely, the rank is determined by the dimension of the smallest face that contains the optimum (e.g. the pairs $(K_\ell, K_\ell^\sigma)$ on the edges of $S$ have rank at most 2 for example, while those on the vertices are rank 1).

We can identify different degrees of sparsity depending on how the rank of the hidden representations scales with the number of datapoints $N$: if the rank is $o(N)$ the covariances $K_\ell, K_\ell^\sigma$ are low-rank (in the traditional linear sense) and for shallow networks the effective number of parameters (i.e. the number of parameters at the start of the plateau) is $o(N^2)$, if the rank is $o(\sqrt{N})$ then for deep networks the effective number of parameters $o(N)$. This could explain why very large networks with 'too many parameters' are able to generalize, since their effective number of parameters is of the order of the number datapoints. Very large networks can therefore be trained safely knowing that thanks to $L_2$-regularization, the network is able to recognize what is the 'right' width of the network.

## 4.2 Tightness of the Upper-Bound

In this section, we construct a pair of input and output datasets $X$ and $Y$, both in $\mathbb{R}^{N \times N}$, such that for any optimal parameters $\mathbf{W}$ of a ReLU network of depth $L = 2$ with no bias ($\beta = 0$), the rank of the hidden representation $\text{Rank}_\sigma(K_1, K_1^\sigma)$ is greater than $N^2/4$.

Note that one can write the decomposition (4.1) as $K = C^T C$ and $K_\sigma = B^T B$ where $C = (z_1, \ldots, z_k)$ and $B = \text{ReLU}(C)$ is obtained by applying elementwise the ReLU to $C$. Key to our construction is the fact that $B$ is then a matrix with non-negative entries: the matrix $K_\sigma$ is completely positive and $\text{Rank}_\sigma(K, K^\sigma)$ can be studied using the CP-rank of $K$:

**Definition 12.** A $N \times N$ matrix $A$ is completely positive if $A = B^T B$ for a $k \times N$ matrix $B$ with non-negative entries. The CP-rank $\text{Rank}_{cp}(A)$ of a completely positive matrix $A$ is the minimal integer $k$ such $A = B^T B$ for a $k \times N$ matrix $B$ with non-negative entries.

When $\sigma$ is the ReLU, the kernel $K_\ell^\sigma$ is completely positive for all hidden layers $\ell$, and thus

$$\max\left(\text{Rank}(K_\ell), \text{Rank}_{cp}(K_\ell^\sigma)\right) \le \text{Rank}_\sigma(K, K^\sigma).$$

In order to obtain the tightness of the upper bound, we proceed in two steps: first, we construct a completely positive matrix $A$ with high CP-rank, and then construct inputs $X$ and outputs $Y$ such that the optimal hidden covariance $K_1 = K_1^\sigma$ for a depth $L = 2$ network equals the matrix $A$.

As shown in [7], bi-partite graphs can be used to construct matrices with high CP-rank. We refine this by showing that graphs on $N$ vertices without cliques of 3 or more vertices lead to $N \times N$ matrices with CP-rank equal to the number of edges, and as a corollary, we construct a completely positive matrix with CP-rank equal to $N^2/4$.

**Proposition 13.** *Given a graph $G$ with $N$ vertices and $k$ edges, consider the $k \times N$ matrix $E$ with entries $E_{ev} = 1$ if the vertex $v$ is an endpoint of the edge $e$ and $E_{ev} = 0$ otherwise. The matrix $A = E^T E$ is completely positive and if the graph $G$ contains no cliques of 3 or more vertices then $\text{Rank}_{cp}(A) = k$.*

Hence, to obtain a completely positive matrix of high CP-rank, it remains to find a graph with no cliques and as many edges as possible. For even $N$, we consider the complete bipartite graph, i.e. the graph with two groups of size $N/2$ and with edges between any two vertices iff they belong to different groups. For this graph, the matrix $B_N = E^T E$ takes the form of a block matrix:

$$B_N = \begin{pmatrix} \frac{N}{2} I_{\frac{N}{2}} & \mathbf{1}_{\frac{N}{2} \times \frac{N}{2}} \\ \mathbf{1}_{\frac{N}{2} \times \frac{N}{2}} & \frac{N}{2} I_{\frac{N}{2}} \end{pmatrix}$$

where $\mathbf{1}_{\frac{N}{2}}$ is the $N/2 \times N/2$ matrix with all ones entries. Since this bipartite graph has no cliques and $N^2/4$ edges, from the previous proposition, we obtain $\text{Rank}_{cp}(B_N) = \frac{N^2}{4}$.

The following proposition shows how for any completely positive matrix (with CP-rank $k$) there is a dataset such that a shallow ReLU network will have a hidden representation pair $(K_1, K_1^\sigma)$ of rank $k$:

**Proposition 14.** *Consider a width-$n$ shallow network ($L = 2$) with ReLU activation, no bias $\beta = 0$, $n_0 = N$, $n_1 \geq N(N+1)$, input dataset $X_N = I_N$, and any output dataset $Y_N$ such that $\left(Y_N^T Y_N\right)^{\frac{1}{2}}$ is a completely positive matrix with CP-rank $k$.*

*At any global minimum of $R_n(X_N, Y_N)$, we have $\mathrm{Rank}_\sigma (K_1, K_1^\sigma) = k$. Furthermore for $\lambda$ small enough, at any global minimum of $\mathcal{L}_{\lambda,n}^{\mathrm{MSE}}(\mathbf{W}) = \frac{1}{N} \|Y(X_N; \mathbf{W}) - Y_N\|_F^2 + \lambda \|\mathbf{W}\|^2$, we have $\mathrm{Rank}_\sigma (K_1, K_1^\sigma) \geq k$.*

By Proposition 14, with the outputs $Y_N = B_N$, the rank of the hidden representations (and the start the plateau) is larger or equal to $\frac{N^2}{4}$. This shows that the order $N^2$ of the bound of Lemma 10 is tight when it comes to data-agnostic bounds. However under certain assumptions on the data one can guarantee a much earlier plateau.

For example, if we instead apply Proposition 14 to a task closer to classification, where the columns of the outputs $Y_N \in \mathbb{R}^{n_L \times N}$ are one-hot vectors, then $(Y_N^T Y_N)^{\frac{1}{2}}$ is (up to permutations of the columns/lines) a block diagonal matrix with $n_L$ constant positive blocks, which is completely positive with CP-rank equal to the number of classes $n_L$. This is in line with our empirical experiments in Figure 2 where we observe in MNIST a plateau starting roughly at a width of 10, which is the number of classes.

Another example where the structure of the data leads to an earlier plateau is when the input and output dimensions are both 1, in which case we can guarantee that the start of the plateau grows at most linearly with the number of datapoints $N$:

**Proposition 15.** *Consider shallow networks ($L = 2$) with scalar inputs and outputs ($n_0 = n_2 = 1$), a ReLU nonlinearity, and a dataset $X, Y \in \mathbb{R}^{1 \times N}$. Both the representation cost $R_n(X, Y)$ and global minimum $\min_{\mathbf{W}} \mathcal{L}_{\lambda,n}(\mathbf{W})$ for any $\lambda > 0$ are independent of the width $n_1$ as long as $n_1 \geq 4N$.*

More generally, we propose to view the start of the plateau as an indicator of how well a certain task is adapted to a DNN architecture. An early plateau suggests that the network is able to solve the task optimally with very few neurons, in contrast to a late plateau. The fact that the optimal network requires few neurons (and hence few parameters) can be used to guarantee good generalization.

## 4.3 Conclusion

We have given two reformulations of the loss of $L_2$-regularized DNNs. The first works for a general non-linearity and shows how the hidden representations of the inputs $Z_1, \ldots, Z_{L-1}$ are learned to interpolate between the input and output representations, as a balance between attraction and repulsion forces for every layer. The second reformulation for homogeneous non-linearities allows us to analyze a sparsity effect of $L_2$-regularized DNNs, where the learned networks are equivalent to another network with much fewer neurons. This effect can be visualized by the appearance of a plateau in the minimal loss as the number of neurons grows, the earlier the plateau, the sparser the solution, since an early plateau means that very few neurons were required to obtain the same loss as a network with an infinite number of neurons. We show that this plateau cannot start later than $N(N+1)$, and then show that the order of this bound is tight by constructing a toy dataset for which the plateau starts at $N^2/4$, however, we observe that on more traditional datasets, the start of the plateau can be much earlier.

## Acknowledgements

C. Hongler acknowledges support from the Blavatnik Family Foundation, the Latsis Foundation, and the NCCR Swissmap.

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
