# OpenReview forum: "Feature Learning in $L_2$-regularized DNNs: Attraction/Repulsion and Sparsity"
_NeurIPS.cc/2022/Conference — NeurIPS 2022 Accept_

### Official Review · Reviewer_LnjM · 2022-07-11

**Rating:** 7
**Confidence:** 3
**Soundness:** 3 good
**Presentation:** 3 good
**Contribution:** 3 good

**Summary:**

This paper studies the minima of the loss of $L_2$ regularized fully-connected DNNs. They show that the loss in terms of the parameters can be reformulated into a loss in terms of the layerwise activations of the training set, which has an attraction/repulsion problem formulation. For positively homogeneous nonlinearities, they show that the loss can be further reformulated in terms of the covariances of the hidden representations, which take the form of a partially convex optimization over a convex cone.

**Questions:**

Some typos:
- Line 124 $\mathcal{L}^r_{\lambda}$, should be $\mathcal{L}_{\lambda}$?
- Line 130, $\mathcal{L}_{\lambda}(Z_1, \dots, Z_L)$ should be $\mathcal{L}^r_{\lambda}(Z_1, \dots, Z_L)$?

**Limitations:**

Yes

**Strengths And Weaknesses:**

Strengths:
- The two reformulations of loss of $L_2$ regularized DNNs are very novel, which may be helpful for our understanding of DNNs.
- The second reformulation has some implications for the sparsity of the local minimum of homogeneous DNNs.

Weaknesses:
- All the theoretical results are limited to the minima of the loss of $L_2$ regularized NN, i.e. based on the assumption that the minima can be achieved. They also assume the minima can perfectly interpolate the dataset.

---

> ### Author Response · Authors · 2022-08-02
> **Response**
>
> Thanks for your review and for identifying the typos, we have corrected them.

---

### Official Review · Reviewer_GKt8 · 2022-07-11

**Rating:** 5
**Confidence:** 3
**Soundness:** 3 good
**Presentation:** 3 good
**Contribution:** 3 good

**Summary:**

This paper provided two reformulations of the L2-regularized DNN using the framework of representation cost. The first reformulation characterizes feature learning process in terms of attraction and repulsion, and further established an interesting connection between training a L2-penalized DNN with a partial convex optimization over a translated convex cone. The second reformulation utilizes the covariance learning and provides a mechanism for learning the sparsity effect of the L2-regularization in training of DNN with homogeneous non-linearities.

**Questions:**

For the second reformulation, it is great to have the understanding of no later than when will we observe the plateau. Could we have any understanding of how early the plateau could occur? I think this might be interesting, especially since an early occurrence of the plateau is observed in many datasets.


**Limitations:**

Please see the comments in the previous 2 sections.

**Strengths And Weaknesses:**

Strengths:

1. The two reformulations are interesting and lead to insightful understanding on the effect of L2 regularization in DNN trainings.

2. The second reformulation establishes an interesting sparsity result for homogeneous DNN (Proposition 7), and the tightness of is further analyzed.

Weakness:

While I enjoy learning these insights and new understandings of the effect of L2 regularization in DNN, I would definitely find it helpful to have more discussions on their implications. Specifically,

1. the first reformulation leads to the attraction-repulsion effects, which could potentially lead to better algorithmic understanding / design. As mentioned in point 3 in the bottom of pp.4, a block (of three layers) cordinate descent type of algorithm could be developed to minimize the multiplicative distances. But how would one deal with the interpolation requirement in the representation costs? While it might be natural to consider a projected gradient descent, going in this direction seems to be a waste of the established decomposition.

2. Proposition 5 establishes the equivalence between training a L2-regularized DNN and a partially convex problem over a translated convex cone. What actual implication could we obtain? What actual benefit can the partial convexity offer?

---

> ### Author Response · Authors · 2022-08-02
> **Response**
>
> Thanks for your thoughtful review. Regarding the weakness you describe:
>  - As mentioned in the paper, we have not analyzed in detail the possibility
> of optimizing either of the reformulations directly. We have added
> a new remark regarding the interpolation requirements: for gradient
> flow, there is no need to do a projection with the first reformulation
> since the loss explodes in the vicinity of points that do not satisfy
> the constraints $P_{\mathrm{Im}Z_{\ell+1}^{T}}\subset P_{\mathrm{Im}\left(Z_{\ell}^{\sigma}\right)^{T}}$:
> this suggests that the gradient descent with a small enough learning
> rate might naturally avoid leaving the constrained region, hence removing
> the need for projections.
>  - The locality property of our reformulations leads to an interesting
> connection to some recent work that tries to develop biologically
> plausible ways to train DNNs (in contrast to backpropagation which
> is not local). Actually, our first reformulation already resembles
> the similarity matching cost from [1]
> which can be optimized in a biologically plausible manner. Our reformulations
> could play a role in the development of biologically plausible training
> techniques which recover the same final parameters as backpropagation
> with $L_{2}$ regularization.
> - The observation that the loss is partialy convex on a convex cone
> was added mostly to help visualize the loss; the only practical implication
> of this property that we are aware of is that, given the representation
> at $\ell-1$ and $\ell+1$, the optimal representation at $\ell$
> is unique.
>
> Regarding your question, Proposition 14 actually gives an exact start
> of the plateau (for the representation cost), furthermore, it actually
> applies to any outputs $Y_{N}$ such that $\left(Y_{N}^{T}Y_{N}\right)^{\frac{1}{2}}$
> is completely positive with CP-rank $k$, in which case the start
> of the plateau is at $k$. Using this updated Proposition 14, one
> can identify datasets with a plateau starting at virtually any possible
> point. For one-hot outputs (which is similar to a classification task)
> the start of the plateau is at the number of classes (which is in
> line with our MNIST experiments).
>
> We have also added a new proposition: for shallow networks with one
> input and one output, the start of the plateau is upper bounded by
> $4N$, leading to a much earlier plateau.
>
> [1] https://proceedings.neurips.cc/paper/2019/file/222afbe0d68c61de60374b96f1d86715-Paper.pdf

---

> > ### Comment · Reviewer_GKt8 · 2022-08-07
> > **Response to the authors**
> >
> > I appreciate the authors' time and effort in addressing my comments. I find the discussions useful.

---

### Official Review · Reviewer_UVbc · 2022-07-17

**Rating:** 5
**Confidence:** 3
**Soundness:** 3 good
**Presentation:** 3 good
**Contribution:** 2 fair

**Summary:**

The authors study deep feedforward neural network training with $\ell^2$
regularization on the weight matrices. They reformulate this cost (via
an equivalent optimization problem) in two different ways, in order to get
insight into the features that are learned at optimality. The first
reformulation expresses the network weights in terms of the preactivations, via
an orthogonal projection trick; the authors interpret the resulting problem in
terms of the local interactions between neighboring preactivations across
layers of the network. The second reformulation builds off the first: it
replaces the preactivations with the corresponding covariance matrices, and the
authors interpret the resulting cost in terms of the rank constraints imposed
by this structure when the layer widths are sufficiently large relative to the
number of samples, which implies some sparsity of the optimal features. Small
experiments are presented to illustrate the interpretations.



**Questions:**

I am not able to understand the claim in line 61 of the appendix that $\Psi$ is
norm-preserving. Can you justify this? It seems like if some features are
poorly conditioned, the norm of the image of $\Psi$ will blow up. I am not sure
about the correspondence between minimizers asserted in Proposition 1 without
this justification.

I was a bit confused about Figure 1 -- are these plots for a network trained on
the reformulated loss (i.e. not really a network -- just learning features), or
a standard optimization over the weights $W$? Are these figures showing the
different attractive/repulsive terms in the trained network, or the initial
network?

Quadratic overparameterization may be more than is necessary for many tasks,
e.g. as discussed in [1], or when data have low-dimensional structure [2].
Can the requirements be relaxed (in principle?) in the latter case? What can
be said theoretically in this framework about networks that are less
overparameterized than quadratic?

[1] https://arxiv.org/abs/2205.10217
[2] https://openreview.net/forum?id=O-6Pm_d_Q-


**Limitations:**

Yes.

**Strengths And Weaknesses:**

## Strengths

- The reformulation in terms of preactivations is insightful, and gives an
  interesting framework for understanding global minimizers of the deep network
  training cost.
- The authors focus on tools that apply to deep networks, which is important --
  many works in the theoretical literature are restricted to shallow networks.

## Weaknesses

- It is slightly unclear what to conclude from the authors' reformulations and
  analysis. The authors discuss the fact that the reformulated optimization
  problems are not computationally useful, given the undesirable scaling with
  the number of data samples relative to training the network weights (section
  3.3); for the second reformulation, there is no correspondence between the
  objective landscapes of the original and reformulated problem (and see a
  question below about the first reformulation). It could be helpful to see
  some experiments on toy data where the scaling is less of an issue to see
  whether or not the reformulated landscapes empirically have some similarities
  to the original landscape, and/or provide useful insights.
- The analysis is focused on understanding learned features in the network, but
  it does not involve any specific structural assumptions on $Y$ and $X$. It
  would be interesting to see if such assumptions can be made, and accordingly
  the conclusions of the second reformulation in Section 4 could be
  strengthened.


## Minor Issues

- line 78: is the 'plus one' in the weight matrix dimensions definition
  misplaced? (the weights multiply the 'superscript sigma' preactivations,
  which are the lifted ones)
- line 112: orthogonal complement
- Proposition 1 proof: there is a typo in the definition of the $\Psi$ map (the
  typo is repeated in other contexts in the proof).  The typo extends to the
  statement of the proposition in the main paper, unless I am missing
  something.  Also in the proof, there are missing $r$ superscripts on some
  losses.
- line 137 (item 2): how are these fractional powers to be interpreted for the
  generally non-square features/input data?
- Proposition 3: the usage of the $\mathrm{cone}$ operator doesn't match the
  way the notation is used above (are the braces and parentheses accidentally
  placed in the wrong order at eqn below line 188 and similar?)

---

> ### Author Response · Authors · 2022-08-02
> **Response**
>
> Thanks for your thorough review.
>
> Regarding the weakness you mention:
> -  As mentioned in the article, we decided to focus on the theoretical
> implications of the reformulations. But there
> is an interesting connection that we have added to the main: the locality
> property of both reformulations could be interesting for developing
> biologically plausible training techniques for DNNs. Actually, our
> first reformulation already resembles the similarity matching cost
> from [1]
> which can be optimized in a biologically plausible way. Our reformulations
> could play a role in the development of biologically plausible training
> techniques that recover the same final parameters as backpropagation
> with $L_{2}$ regularization.
> - We have added two examples of datasets with an earlier plateau to
> complement our data-agnostic bound. First, Proposition 14 actually
> applies to any output $Y_{N}$ such that $\left(Y_{N}^{T}Y_{N}\right)^{\frac{1}{2}}$
> is completely positive with CP-rank $k$, in which case the start
> of the plateau will be at $k$. Using this updated Proposition 14,
> one can identify dataset with a plateau starting at virtually any
> possible point. For one-hot outputs (which is similar to a classification
> task) the start of the plateau is at the number of classes (which
> is in line with our MNIST experiments). Second, for shallow networks
> with one input and one output, the start of the plateau is upper bounded
> by $4N$ (we added a new short proposition describing this).
>
> Thanks for pointing to the typos and other minor issues, we have corrected
> them. Regarding the description of the linear case at line 137: actually,
> it was wrong to give a formula for $Z_{\ell}$ (at an optimum, $Z_{\ell}$
> is only determined up to orthogonal transformation to the left), instead,
> we give a formula for the covariance $Z_{\ell}^{T}Z_{\ell}=X^{T}(X^{-T}Z_{L}^{T}Z_{L}X^{-1})^{\frac{\ell}{L}}X$,
> which interpolates between $X^{T}X$ and $Z_{L}^{T}Z_{L}$, which
> is well-defined for any inputs/outputs.
>
> Regarding your questions:
> - You are right that $\Psi$ is not norm preserving. There was a error in line 61, the norm $ ||Z_i ||^2$ simply needs to be replaced by $\sum_{\ell=1}^{L}|| Z_{i,\ell}(Z_{i,\ell-1}^{\sigma})^{+}||_F^2$.
> Thanks for identifying this error.
> - For Figure 1, the networks are trained in the traditional manner (with
> backpropagation) on the original loss and we plot the attraction and
> repulsion forces at the end of training. We made this clearer in the
> caption.
> - As mentioned above, we added a discussion for datasets with earlier
> plateaus and we added two examples. These are examples of datasets
> where the second reformulation is useful when the width of the network
> is either constant in $N$ (equal to the number of classes) or linear
> in $N$.
>
> [1] https://proceedings.neurips.cc/paper/2019/file/222afbe0d68c61de60374b96f1d86715-Paper.pdf

---

> > ### Comment · Reviewer_UVbc · 2022-08-04
> > **further question**
> >
> > Dear authors,
> >
> > Thank you for your response to my review, and for your work in updating the submission and appendices.
> >
> > - I am still not quite following the proof of the claim about minima in Proposition 1. I am fixating on this point because I think the correctness of this claim would raise my appraisal of the submission, since this implies a useful correspondence between the problems' landscapes. The error you have fixed helps my understanding of the claim -- but I am still not following the claim in Line 60 of the appendix, as the term involving the $Z$s after the first equality does not match the structure of the $Z$'s in the definition of $\mathcal{L}^r_{\lambda}$ below line 33, where one has $Z_{\ell} ( Z_{\ell}^\sigma)^+$ rather than $Z_{\ell} ( Z_{\ell-1}^\sigma)^+$. Could you clarify what the argument is that gives the first inequality in line 60? I am assuming here that it is also necessary that $C \geq 0$. If there are other typos, can you please fix these so that the claim can be appreciated and understood?
> >
> > I will respond to the other points you have raised after understanding this point -- thank you.

---

> > > ### Author Response · Authors · 2022-08-05
> > > **Response**
> > >
> > > We apologize, this was another typo, it should read $Z_\ell (Z_{\ell-1}^\sigma)^+$ everywhere (all the typos where $Z_\ell (Z_{\ell}^\sigma)^+$ appears can be replaced by $Z_\ell (Z_{\ell-1}^\sigma)^+$ in the paper).
> > >
> > > The argument for the inequality in line 60 goes as follows:
> > > - Equality: since $ W_i=\Psi (Z_i) $ one has $ \||W_{i,\ell}\||^2_F = \||Z_{i,\ell} (Z_{i,\ell-1}^\sigma)^+ \||^2_F $.
> > > - Inequality: this should be obvious without the typo in the definition the loss $\mathcal{L}^r_\lambda$, as long as the cost $C$ is positive (as you assumed). Note that the assumption $C\geq 0$ is actually not necessary, only that it is lower bounded $C \geq c$: we would then get the same upper bound up to a constant $-\frac c \lambda$ which is still fine since we only need boundedness.
> > > - Strict inequality: this is by assumption.

---

> > > > ### Comment · Reviewer_UVbc · 2022-08-09
> > > > **response**
> > > >
> > > > Dear authors,
> > > >
> > > > Thanks for the extensive clarifications. I checked the proofs again and am happy with the claims asserted. A couple of additional comments:
> > > > - In Proposition 1, is it being assumed here that $\min_{\ell} n_{\ell} \geq N$? If I am understanding correctly, I think this is necessary to have $\Phi \circ \Psi = \mathrm{Id}$ (i.e. because you want $(Z_{\ell-1}^\sigma)^+ Z_{\ell-1}^\sigma = I$). Perhaps it is possible to include a more detailed argument here to remove this hypothesis, as you do when considering $\Psi \circ \Phi$ in the second part of the proof?
> > > > - I would recommend giving this proof (and perhaps others that I did not read line-by-line) a careful read for typos / outdated notation (e.g. $Y_L$), as it is fairly sparse on details and as a result hard to read -- the existence of typos make it impractical to parse.
> > > > - Can this proof be extended to saddle points, as the paragraph header in the proof indicates the authors intended to? I see that the proof only uses the topological notion of local minimizer -- for saddles perhaps some differentiability needs to be assumed?
> > > >
> > > > After reviewing the revisions and the comments of the other referees, I am increasing my score. I think the paper presents interesting observations and correct technical arguments, although the general implications of the study are not completely clear.

---

> > > > > ### Author Response · Authors · 2022-08-10
> > > > > **Response**
> > > > >
> > > > > Thanks for the discussions and for updating your score.
> > > > >
> > > > > Regarding your questions:
> > > > > - We do not assume that $n_\ell \geq N$ in Proposition 1. Where in the proof do you think $Z_\ell^\sigma (Z_\ell^\sigma)^+=I$ is needed?
> > > > > - We already did that since your last response but we couldn't upload a new version. Sorry again for the typos.
> > > > > - That's a very interesting question that I investigated some time ago. From what I remember one first needs to define a gradient for the first reformulation (it is not differentiable but there is some notion of gradient which makes the most sense). Under this definition you get an equivalence between critical points and therefore between saddles (since saddles are critical points that are not local minimas nor local maxima and there are no local maximas). We decided to not put this in the main since it requires defining this notion of gradient and justifying it, but we could add this discussion to the appendix.

---

### Meta-Review · Area_Chair_Dspf · 2022-08-26

**Recommendation:** Accept
**Confidence:** Less certain

**Metareview:**

This paper studies l_2 regularization on the norms of fully connected networks and discusses how that influences feature learning. It gives two reformulations of the loss (from parameters to the activations) which give intuition on attraction/repulsion effects and an "effective" number of neurons (that the minimum can always be achieved by a network of size N(N+1)). Overall the reviewers find the reformulations novel and interesting, while there are some concerns most are addressed in the author response.

**Award:**

No

---

### Decision · Program_Chairs · 2022-09-14

Accept